# Effect of Low Intensity Pulsed Ultrasound (LIPUS) on Tooth Movement and Root Resorption: A Prospective Multi-Center Randomized Controlled Trial

**DOI:** 10.3390/jcm9030804

**Published:** 2020-03-16

**Authors:** Tarek El-Bialy, Khaled Farouk, Terry D. Carlyle, William Wiltshire, Robert Drummond, Tim Dumore, Kevin Knowlton, Bryan Tompson

**Affiliations:** 1Division of Orthodontics, Katz Group Centre for Pharmacy and Health Research, School of Dentistry, University of Alberta, Edmonton, AB T6G 1C9, Canada; 2Department of Orthodontics, Faculty of Dental Medicine, Al-Azhar University, Cairo, Egypt; 3Division of Orthodontics, School of Dentistry, Faculty of Medicine and Dentistry, University of Alberta, Edmonton, AB T6G 1C9, Canada; khaledfarouk@azhar.edu.eg; 4Division of Orthodontics, Graduate Orthodontic Program, Faculty of Medicine and Dentistry, University of Alberta, Edmonton, AB T6G 1C9, Canada; terry.carlyle@str8teeth.com; 5Department of Preventive Dental Science (incl. Community Dentistry, Orthodontics and Pediatric Dentistry), Division of Orthodontics, Graduate Orthodontic Program, College of Dentistry, University of Manitoba, Winnipeg, MB R3E 0W3, Canada; wa.wiltshire@umanitoba.ca; 6Division of Orthodontics, Graduate Orthodontic Program, College of Dentistry, University of Manitoba, Winnipeg, MB R3E 0W3, Canada; robert.drummond@umanitoba.ca (R.D.); tim@drdumore.com (T.D.); 7Graduate Orthodontic Program, University of Toronto, Toronto, ON M5G 1G6, Canada; 8Discipline Head Orthodontics, Faculty of Dentistry, University of Toronto; Division of Orthodontics, Hospital for Sick Children, University of Toronto, Toronto, ON, M5G 1G6, Canada; bryan.tompson@dentistry.utoronto.ca

**Keywords:** orthodontic tooth movement, acceleration, low-intensity pulsed ultrasound, LIPUS, clinical trial

## Abstract

The aim of this study was to evaluate the possible effect of low intensity pulsed ultrasound (LIPUS) on tooth movement and root resorption in orthodontic patients. Twenty-one patients were included in a split-mouth study design (group 1). Ten additional patients were included with no LIPUS device being used and this group was used as the negative control group (group 2). Group 1 patients were given LIPUS devices that were randomly assigned to right or left side on upper or lower arches. LIPUS was applied to the assigned side that was obtained by randomization, using transducers that produce ultrasound with a pulse frequency of 1.5 MHz, a pulse repetition rate of 1 kHz, and average output intensity of 30 mW/cm^2^. Cone-beam computed tomography (CBCT) images were taken before and after treatment. The extraction space dimensions were measured every four weeks and root lengths of canines were measured before and after treatment. The data were analyzed using paired *t*-test. The study outcome showed that the mean rate of tooth movement in LIPUS side was 0.266 ± 0.092 mm/week and on the control side was 0.232 ± 0.085 mm/week and the difference was statistically significant. LIPUS increased the rate of tooth movement by an average of 29%. For orthodontic root resorption, the LIPUS side (0.0092 ± 0.022 mm/week) showed a statistically significant decrease as compared to control side (0.0223 ± 0.022 mm/week). The LIPUS application accelerated tooth movement and minimized orthodontically induced tooth root resorption at the same time.

## 1. Introduction

Facial and dental aesthetics play a very important part of the social measurement in the overall attractiveness. A person with a pleasing smile and facial appearance is more expected to encounter favorable appraisals and assessments from his/her peer-group, seniors, and employers [1]. On the other hand, patients with malocclusion may have issues with their oral, physical, and psychological health, depending on the severity of the malocclusion [2,3]. The psychological aspect is the main driving force for a person to undergo orthodontic treatment, as well as the quintessential aspects to adhere to the treatment. A study by Sergl and Zentner [4] on the psychological side of the orthodontic patients found that about two-thirds were concerned about poor aesthetic. Nevertheless, the overall duration of orthodontic treatment not only worries the patients and parents, but also the orthodontists [5]. 

An increasing number of adults are undergoing orthodontic treatment and the most common concern for these patients is the treatment duration, in addition to the type of orthodontic appliance to be used in the treatment [5]. The average orthodontic treatment is from 2–3 years [6,7]. Prolonged orthodontic treatment not only affects the patients’ compliance [8], but also predisposes the patient to higher caries incidence, white spot lesions [9], gingival inflammation and recession, and root resorption [10]. 

Hence, accelerating orthodontic treatment with preservation of the integrity of tooth structure and alveolar bone has received increasing attention by orthodontic academics and clinicians. Various procedures have been employed to reduce the treatment duration. These techniques include low-level laser therapy [11], pulsed electromagnetic fields [12], electrical currents [13], corticotomy [14], distraction osteogenesis [15], mechanical vibration [16], and photobiomodulation [17]. However, there is no confirmed data that these techniques can minimize orthodontically induced tooth root resorption (OITRR), while at the same time accelerating tooth movement. 

Orthodontic tooth movement (OTM) is a complex inflammatory process that depends on the remodeling of alveolar bone that surrounds the tooth. This process comprises a cascade of events involving the secretion of biochemical mediators e.g., cytokines (interleukin IL-1. IL-2, IL-6, etc.), growth factors (transforming growth factor-β (TGF-β), fibroblast growth factor (FGF), bone morphogenetic protein (BMP), and vascular endothelial growth factor (VEGF)) and prostaglandins (PG-E); the recruitment of osteoblasts and osteoclasts. The rate-limiting factor of OTM is the bone resorption [18,19]. One of the common side effects of orthodontic treatment is OITRR and the second most common root resorption after pulp infection-related root resorption [20]. The molecular mechanism of OITRR is still unknown; however, it has led to many malpractice lawsuits against orthodontists [21,22]. The prevalence of root resorption increased up to 73%–80% following orthodontic treatment [23,24]. 

Recently, low-intensity pulsed ultrasound (LIPUS) has been shown to increase the rate of tooth movement in ex-vivo mandible slice organ culture and in animals and decrease OITRR [25,26]. Ultrasound is a form of mechanical energy that can be transmitted through different tissues as pressure waves. The frequency of these waves is above the limit of human hearing. Ultrasound has been used in medicine in surgical operation, therapeutic application, and diagnostic [27,28]. It has also been reported that LIPUS accelerates normal fracture repair when applied daily for a period of three weeks when applied for 15 to 20 min per day with intensity between 30 to 50 mW/cm^2^ [29]. A recent clinical study in human patients showed that the intermittent use of LIPUS (at days 0, 3, 5, 7, 14, and every 15 days afterward) had an increased rate of tooth movement, however the LIPUS apparatus used was extraorally applied and the device was applied by the operator to one side of the mouth [30]. It has been reported that the stimulatory effect of LIPUS is dose dependent [31]. It is not known whether the application of LIPUS intraorally and on a daily basis would be beneficial in accelerating tooth movement or not. 

The primary aim of the current study was to evaluate the effect of LIPUS on the rate of orthodontic tooth movement in the split-mouth clinical trial. The null hypotheses were: (1) there is no significant difference in the rate of tooth movement between LIPUS and control side; and, (2) there is no significant difference in the root resorption after orthodontic tooth movement between LIPUS and control sides. The alternate hypotheses are (1) the LIPUS treated side will have accelerated tooth movement and reduced root resorption when compared to the control side.

## 2. Experimental Section

The study was a split-mouth randomized controlled trial with a 1:1 allocation ratio. This was a multicenter study that was conducted across five sites—three public universities and two private clinics. The respective ethics committee at each participating institute approved the study. The participants selected were acquainted with the study procedures and radiation exposures, and then informed written consents were signed. This study is also registered at the ClinicalTrials.gov with the identifier number: NCT01828164. 

Subjects who met all the following criteria qualified for entry into the study:permanent dentition and ages between 12 and 40 years old;first premolar extractions indicated for correcting the existing dental malocclusion (to eliminate overjet or crowding) with a minimum of 3mm of extraction space;good oral hygiene and compliance; and,no history of systemic disease.

The exclusion criteria were:any compromised medical or dental condition that prevents the subject from participating in the trial or using a medical device (like diabetes, renal failure, under corticosteroid treatment, etc.);any implanted assistive device e.g., pacemakers, cochlear implants, etc.;chronic use of medications affecting orthodontic tooth movement e.g., Bisphosphonate; and,pregnant females.

### 2.1. Sample Size Calculation

Based on the previous studies, the average orthodontic tooth movement for canine retraction in human subjects is 1.11 mm per month with a standard deviation of 0.43 mm [32,33]. In the previous dog study [26], the orthodontic tooth movement on the side receiving LIPUS treatment was approximately 50% faster when compared to the control side. This acceleration was obtained for dogs being treated for 20 min. every day (100% compliance). Hence, the expected mean monthly rate of tooth movement was 1.66 mm for the LIPUS treated side + orthodontic braces group and 1.1 mm for the orthodontic braces alone group. Based on an analysis that was performed using the paired *t*-test, a minimum of 10 extraction sites per group (with 100% usage compliance) were required, for power of 80% and at a significance level of 0.05. Figure 1 presents details of the recruited patients and reasons for dropping out of the trial. In short, sixty patients were screened for this clinical trial and forty-seven met the selection criteria. From these forty-seven patients, twenty-one patients completed the clinical trial with a per protocol device usage compliance of ≥ 67%. 

### 2.2. Ethics Approval and Consent to the Patients

The study was approved by the Research Ethics Committee of University of Alberta (PR-0018 approval date 27 September 2012), University of Toronto (PR-0031 approval date 8 January 2013), Strathcona Orthodontics (PR-0018.3 approval date 25 March 2014), University of Manitoba (PR-0041 approval date 26 March 2014) and Dr. Dumore and Team Orthodontics (PR-0041 approval date 26 March 2014). All the procedures complied with the Code of Ethics of the World Medical Association (Declaration of Helsinki), Canadian Medical Device Regulation SOR-98-282, ICH tripartite guidelines for Good Clinical Practice and ISO 14155:2011 Clinical investigation of medical devices for human subjects. The participants selected were acquainted with the study procedures and radiation exposures, then informed written consents were signed.

### 2.3. Randomization

The prospective patients were randomized into two groups. Group 1, in the split-mouth study, included twenty-one patients who fulfilled the pre-approved selection criteria and completed their clinical trial participation (24 weeks or until the closure of the extraction space on either side, whichever period was shorter). This involved one side of the patient’s mouth receiving LIPUS treatment, while the other side of the same patient was inactivated and served as a placebo or positive control. LIPUS device was provided with the active treatment zones of the mouthpiece pre-selected by the manufacturer (SmileSonica Inc., Edmonton, AB, Canada). The device was provided free of cost to the patients that were enrolled in the study. The active side of the mouthpiece was done on a per-site basis according to a predetermined randomized allocation sequence that was created by an independent third party. Patient and investigator were both blinded to which side was active or inactive. Blinding was satisfied since LIPUS cannot be heard or felt. The study research assistant or principal investigator took the intra-oral measurement, on a cast or scan measurements at the beginning of canine retraction, and every four weeks. All of the clinical measurements were made by the same person at each site to ensure consistency of the measurement consistency and eliminate measurement errors. 

The split-mouth design model was selected for this clinical trial, because the right and left sides of a dental arch are normally quasi-symmetric with similar properties and dimensions (e.g., tooth size, gum thickness, etc.). However, another group was also used due to the probability of transmission of the ultrasound waves from the active side to the passive side (cross mouth contamination). Group 2 was composed of 10 patients that had no LIPUS treatment on either side, which served as a negative control group and was included in the study (no blindness was applied to this group). 

### 2.4. LIPUS Device

LIPUS was applied for 20 min. per day while using a custom-made ultrasound device for the duration of trial i.e., 24 weeks or until the closure of the extraction space on either side, whichever period was shorter. The device had a mouthpiece that was similar to a mouthguard connected to the handheld electronics, which had a screen that provided information regarding the treatment (Figure 2). The transducers are embedded in the mouthpiece that was located at the tooth root level. The coupling gel was supplied by the manufacturer, which was applied to the inside of the mouthpiece before the start of each treatment, so that the LIPUS can be properly transmitted from the mouthpiece through the gums to the teeth roots. The LIPUS device had an internal memory microchip that recorded the date, time, and duration of each LIPUS application. The LIPUS output consisted of ultrasound with a frequency of 1.5 MHz, a pulse repetition rate of 1 kHz, and an average output intensity of 30 mW/cm^2^. Each patient had a twin device that was exchanged every month, so that the device could be returned to the manufacturer to verify the calibration of LIPUS output intensity. 

### 2.5. Intervention

All of the patients underwent first premolars extractions and subsequent canine retraction using a 0.017” × 0.025” Titanium-Molybdenum alloy (TMA) canine retraction T-loop that was constructed for each extraction site, as described previously by Burstone (1962) [34]. The T-loop was activated by cinching the posterior leg of the spring 5–6 mm out of the auxiliary tube of the first molar. The spring was reactivated every two months (Figure 3). Anchorage was done using the second premolar, first and second molars utilizing transpalatal and lingual arch in maxilla and mandible arch, respectively. A pre-retraction (T_0_) alginate impression and CBCT were recorded on the first day of treatment. The follow-up visits were scheduled every four weeks. The intra-oral extraction space was measured at the beginning of the treatment and before the extraction gap closed to calculate a weekly tooth movement rate. Interim visit measurements were not required for determining the overall tooth movement rate during the clinical investigation. The records included pre-treatment, either a large field-of-view CBCT scan or regular cephalometric in addition to a small field-of-view CBCT (the smallest picture possible is taken about 8 cm × 8 cm, 0.3 mm voxel size) (iCAT, Hatfield, PA, USA). 

### 2.6. Outcomes Assessment

#### 2.6.1. Subject Compliance

A computer program that downloaded the usage data from the device at each visit monitored the compliance of the device use. A compliance level of 67% for the duration of the trial was the minimum accepted allowance in any visit. 

#### 2.6.2. Primary Outcome

Three types of tooth movements can occur during orthodontic treatment: pure translation, pure tipping, or a combination of translation and tipping. Only the same type of movement was compared since the clinical study involved comparing the gap closure rates on the two sides of the same arch. If the two sides had different types of tooth movement i.e., one side had a tipping and the other side had a translation, then the data were excluded from the analysis. Measuring the movement of the crowns and root tips for the canines relative to the first molars on the CBCT scans or dental radiographs quantified the type of tooth movement. The amount of tooth movement was calculated by measuring the amount of movement of the canine’s pulp horn relative to the first molar’s pulp chamber and the amount of tooth movement of the canine’s root apex relative to the first molar’s mesial root apex. 

d_Crown_: the amount of tooth movement of the canine’s crownd_Root_: the amount of tooth movement of the canine’s root apex

Tipping = d_Crown_ − d_Root_
(1)$$\mathrm{Translation}=\{\begin{array}{c} d_{\mathrm{Crown}},\left|d_{\mathrm{Crown}}\right|\geq \left|d_{\mathrm{Root}}\right|\\ d_{\mathrm{Root}},\left|d_{\mathrm{Crown}}\right|<\left|d_{\mathrm{Root}}\right|\\ 0,\mathrm{sgn}(d_{\mathrm{Crown}})\neq \mathrm{sgn}(d_{\mathrm{Root}}) \end{array}$$

The PIs or the research assistants took dental cast measurement (or intra-oral measurement if dental casts were not available) at the beginning of canine retraction, and every four weeks to assess the total space closure. The rate of tooth movement was calculated, as follows:d_t0_: the first extraction space measurement (from canine cusp tip to the mesiobuccal groove of the first molar);d_t1_: the last extraction space measurement prior to extraction space closure (i.e., extraction space = 0 mm) on either side; and,t: the number of weeks between d_t0_ and d_t1_.(2)$$\mathrm{Tooth}\;\mathrm{movement}\;\mathrm{rate}=\frac{d_{t0}-d_{t1}}{t}$$

The percentage change was calculated as follows, using:r_Ultrasound_: Ultrasound tooth movement rate; and,r_Control_: Control tooth movement rate.
(3)$$\mathrm{Percent}\;\mathrm{Change}=\frac{\left(r_{\mathrm{Ultrasound}\;}-r_{\mathrm{Control}}\right)}{\left|r_{\mathrm{Control}}\right|}\times 100\%$$

#### 2.6.3. Secondary Outcome: Root Resorption Using CBCT

On the CBCT images, the canine length was calculated from the pulp horn to the root apex, the pulp was found to be more accurate than the cusp tip, because it is not affected by external factors, such as grinding, fillings, etc. Besides, the pulp horn landmark can be precisely identified on the CBCT images [35]. The root apex and the pulp horn of the canine were identified in the axial, sagittal, and coronal sections. The three-dimensional (3D) coordinates of each point were obtained, and the straight-line distance between the apex and the pulp horn was calculated while using the Euclidean distance formula: (4)$$D=\sqrt{({\left(\mathrm{XC}-\mathrm{XR}\right)}^{2}+{\left(\mathrm{YC}-\mathrm{YR}\right)}^{2}+{\left(\mathrm{ZC}-\mathrm{ZR}\right)}^{2}}$$
where D is the tooth length, X is the transversal position (relation to the x-axis), Y is the anteroposterior position (relation to the y-axis), Z is the vertical position (relation to the z-axis), C is the pulp horn, and R is the root apex. 

The root resorption rate was calculated as follows, using:l_(pre-trial)_: the pre-trial tooth length;l_(post-trial)_: the post-trial tooth length; and,t: the number of weeks between l_(pre-trial)_ and l_(post-trial)_.
(5)$$\mathrm{Root}\;\mathrm{resorption}\;\mathrm{rate}=\frac{l_{\mathrm{pre}-\mathrm{trial}}-l_{\mathrm{post}-\mathrm{trial}}}{t}$$

The percentage change was calculated as follows, using:r _Ultrasound_: Ultrasound root resorption rate; and,r _Control_: Control root resorption rate.
(6)$$\mathrm{Percent}\;\mathrm{Change}=\frac{\left(r_{\mathrm{Control}}-r_{\mathrm{Ultrasound}}\right)}{\left|r_{\mathrm{Ultrasound}\;}\right|}\times 100\%$$

### 2.7. Statistical Analysis

Statistical analysis was performed while using the SPSS program version 20 (SPSS Inc, Chicago, IL, USA). The Kolmogorov–Smirnov test was used to test the normality of data distribution, which revealed normal distribution; therefore, parametric tests were used. An unpaired *t*-test was used to compare the differences in the rate of tooth movement and root lengths as an estimation of root resorption between the positive control (placebo) side in the split-mouth group and the negative control results. The paired *t*-test was used to compare the variables between the two sides of the split-mouth group. The significance level (alpha) was set at 0.05.

## 3. Results

The CONSORT flow diagram depicted in Figure 1 provides a summary of subject accountability in the clinical investigation. In summary, out of sixty patients enrolled in the active group, 13 patients did not meet the inclusion criteria after enrollment; hence, out of sixty, only 47 entered the participant flow. Twenty-four of the 47 patients had lower compliance for the device; hence, they were removed from the final analysis. One patient had a non-functional device and for one patient the pre-trial CBCT scan was not taken, hence these two patients were also removed from the final analysis. For the final analysis, there were 21 patients in the clinical trial analysis. The average age of 21 patients was 19.7 ± 6.63 (minimum = 12 years, and maximum = 37 years five months), with five male and 16 female subjects. 

Data from 10 patients were collected and analyzed as a negative control group and they were only compared to the positive control group to study if LIPUS was reaching to the positive control in the active group and were not compared to the active side analysis. On comparing the tooth movement and root resorption between the negative control and positive control group, there was no significant difference between these groups (*p* = 0.11; and *p* = 0.32, respectively), no cross-mouth effect could be detected; hence, the split-mouth model is validated in our study (Table 1).

### 3.1. Tooth Movement Analysis 

In the active treatment group, four patients were excluded due to the type of tooth movement being non-comparable; hence, seventeen patients were included for the final analysis of tooth movement. The mean rate of tooth movement for the LIPUS side was 0.266 ± 0.0927 mm/week and the positive control side was 0.232 ± 0.0855 mm/week (Figure 4). Table 2 shows a comparison of the rate of tooth movement in the LIPUS and positive control side in the active treatment group. The difference was statistically significant (*p* < 0.05). The percentage change of tooth movement rate for each individual patient was first calculated, as this was a split mouth study. Subsequently, the mean of the individual percentage changes for the seventeen patients was calculated, and it resulted in a 29% mean percent increase in the tooth movement rate when compared to the positive control.

### 3.2. Root Resorption Analysis

Six subjects from one trial site had dental radiographs instead of CBCTs (due to the lack of CBCT equipment), and, in two additional subjects, the tip of the canine roots were unintentionally not captured in the CBCT scans; hence, these eight patients were excluded from the root resorption analysis. Consequently, for root resorption analysis, thirteen patients’ split mouth data was included in the final analysis. The mean root resorption rate for the LIPUS side was 0.0092 ± 0.0226 mm/week and for the positive control side 0.0241 ± 0.0223 mm/week (Figure 5). Table 3 shows the descriptive analysis of the LIPUS and the positive control side in the active treatment group. The difference was considered statistically significant (*p* < 0.05).

## 4. Discussion

Apart from the cost of treatment, the second most frequently asked question to the orthodontist is the duration of treatment. Precisely predicting the duration and completing the treatment in the predicted time are important, as they will not only affect the patient’s compliance but will increase recognition for the orthodontist in longer run [36,37]. Especially in increasing number of adult patients undergoing orthodontic treatment, various methods are used in the practice to accelerate the tooth movement and shorten the duration of treatment. These included both invasive and non-invasive methods. Corticotomy was one of the first methods introduced by Kole in 1959 by cutting the alveolar bone around the teeth [38]. LIPUS is an emerging dental technique for accelerating tooth movement, and it has been used in the medical field as a diagnostic, operative, and therapeutic tool for over five decades [27]. 

This clinical investigation aimed to evaluate the effect of LIPUS on the rate of tooth movement and root length changes, as an indication of OITRR. A split-mouth design was chosen in the current study, where LIPUS was randomly allocated to one side of each individual, to minimize the variability between individuals or sites. Patient and principal investigators were both blinded to which side was active/placebo throughout the study. A negative control group was used in this study to overcome the possibility of cross-contamination between LIPUS and the positive control side in each patient. There was no significant difference between the positive and negative controls regarding the rate of tooth movement (*p* = 0.11) and root resorption (*p* = 0.32). This indicates that any LIPUS that might have reached the control side from the treatment side did not have any effect on tooth movement or root resorption minimization. However, it has been reported that LIPUS power attenuates exponentially as it propagates through dentoalveolar structure. 

Twenty-one patients completed the study based on the predetermined selection criteria and device usage compliance. For the tooth movement analysis, there were seventeen split mouth patients, which comprised of six maxilla split mouth patients and eleven split mouth mandible patients. For root resorption analysis, thirteen split mouth patients that were included in the final analysis comprised of four maxilla split mouth patients and nine mandible split mouth patients. Segmented arch mechanics were used to alleviate any possible wire/bracket friction on rate of tooth movement that can be a confounding factor. In the present study, the intraoral LIPUS system provides a statistically significant increase in the tooth movement rate, with an average percentage increase of 29% in tooth movement rate as compared to the control. Similar results were reported in the recent study that was conducted by Maurya et al [30], where LIPUS application increased the orthodontic tooth movement in bimaxillary protrusion cases.

OTM in response to external mechanical forces applied by wire and braces is best explained by “pressure—tension theory”. The direction towards which the tooth moves is the pressure side, while the opposite side is the tension side. On the pressure side, the application of force triggers several metabolic changes in the periodontal ligament (PDL) area, causing inflammation by constricting the blood vessels, causing a lack of nutrient and subsequent hyalinization and cell death [39]. The rate of orthodontic tooth movement depends on the remodeling of the alveolar bone and the rate-limiting factor for tooth movement is bone resorption at the bone and PDL interface [19,40]. Macrophages, odontoclast, and osteoclasts are recruited in the area of hyalinization to eliminate the necrotic tissue, which further leads to resorption of mineralized tissue by secreting tartrate-resistant acid phosphate (TRAP), Cathepsin-K, and matrix metalloproteinase-9 (MMP-9) [41,42].

LIPUS has been shown to have a bio-stimulatory effect on osteoblast and osteoclasts. Additionally, it has shown that LIPUS increases the cell number and activities within PDL that could be important in alveolar bone remodeling [26,43]. The mechanical stimulation from LIPUS is received by the receptors on the cell membrane like integrins [44,45] and GPCR (G-protein coupled receptors) [46] to activate different mechanotransduction pathways in the bone cells. This leads to increased gene expression [47,48], which in turn leads to increased protein expression [49]. LIPUS increases RANK-L (receptor activator of nuclear factor kappa-ligand) protein expression in the osteoclasts to accelerate bone resorption [50,51], while in osteoblasts LIPUS increases bone-forming proteins RUNX2 (runt-related transcription factor 2) [52], OPG (osteoprotegerin) [53], and ALP (alkaline phosphatase) [54]. 

The statistically significant decrease in root resorption with LIPUS application is consistent with previous studies [55]. This could be due to the suppressive effect of LIPUS on cementoclastogenesis [53], alteration in the expression of OPG/RANKL during the orthodontic tooth movement [56], and enhancing tissue regeneration, hence promoting periodontal healing [57]. CBCT is the only radiograph that can evaluate tooth movement and root length in three dimensions.

To the best of our knowledge, this is the first split-mouth design clinical trial reported in literature, where LIPUS’ stimulatory effect on the rate of tooth movement and OITRR was studied. In addition, to the best of our knowledge, there is no other technique than LIPUS that can minimize OITRR and enhance tooth movement at the same time. Future research might be directed to optimize LIPUS output for the possible enhancement of tooth movement and OITRR results. 

The primary limitation of this study is the sample size of twenty-one patients in a split-mouth design (twenty-one data pairs). Secondly, of the twenty-one patients, there were sixteen females and five males. The frequency of female patients in orthodontic practice is more, as females are more concerned about their dental appearance than males [58,59]. A study by Ashari et al [60] also showed about twice the number of females when compared to the male patients. In the future, a clinical trial with a larger sample size, equal gender distribution, and clean treatment and control groups will be undertaken. The present study will help in sample size calculation from the standard deviation calculated and the device usage compliance. 

## 5. Conclusions

Nevertheless, this study should be considered to have important practical implications regarding orthodontic treatment despite the above limitations. LIPUS increased the rate of tooth movement and decreased orthodontically induce root resorption when applied for 20 min per day for up to six months. Based on the results of this study, we reject the null hypothesis and accept the alternate hypothesis that the LIPUS treated side had accelerated tooth movement and reduced root resorption when compared to the control side.

## Figures and Tables

**Figure 1 jcm-09-00804-f001:**
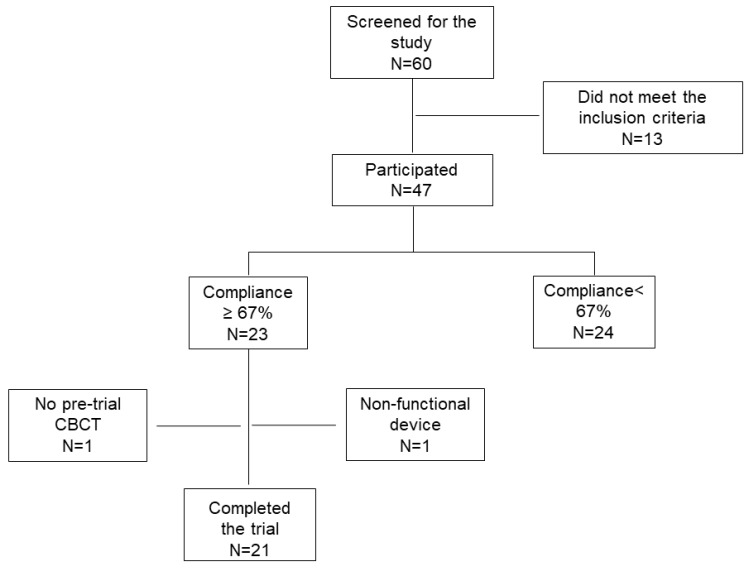
The CONSORT flow diagram.

**Figure 2 jcm-09-00804-f002:**
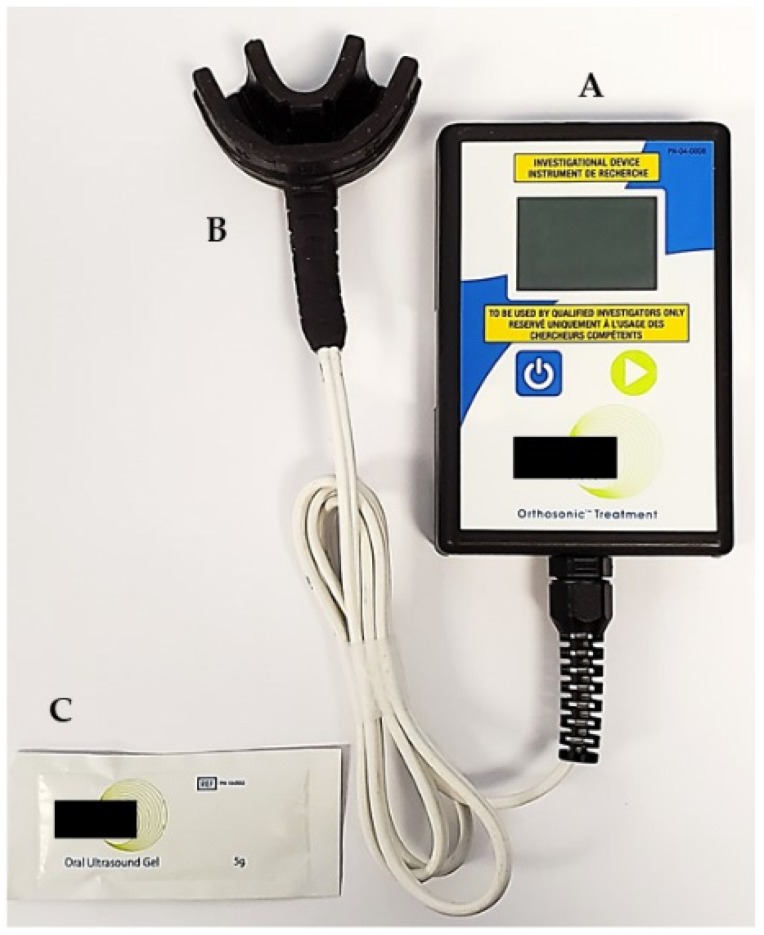
Low-intensity pulsed ultrasound (LIPUS) device used in the clinical trial including **A**: handheld electronics; **B**: mouthpiece containing LIPUS transducers; and, **C**: oral ultrasound gel.

**Figure 3 jcm-09-00804-f003:**
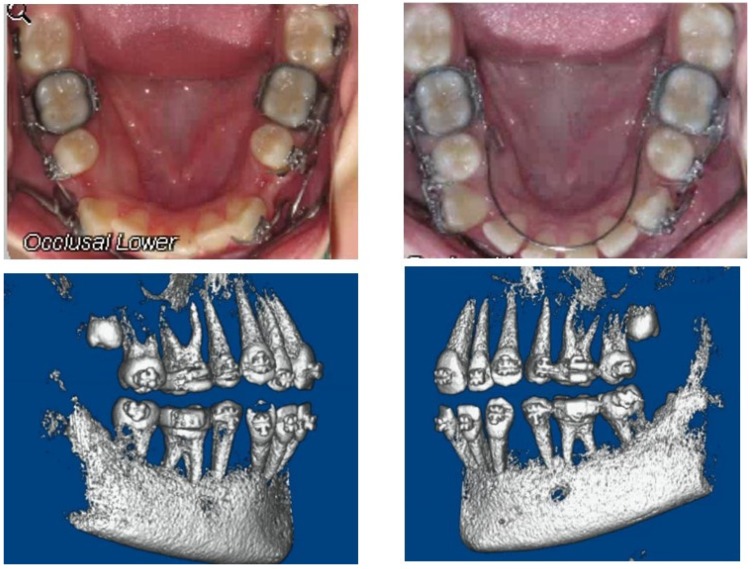
A case in the LIPUS group at the beginning (top left) and two months of the treatment (top right) the patient’s lower right side was treated by LIPUS and lower left side was control. CBCT scan showing root angulation after canine retraction (bottom left LIPUS treated side and right is control).

**Figure 4 jcm-09-00804-f004:**
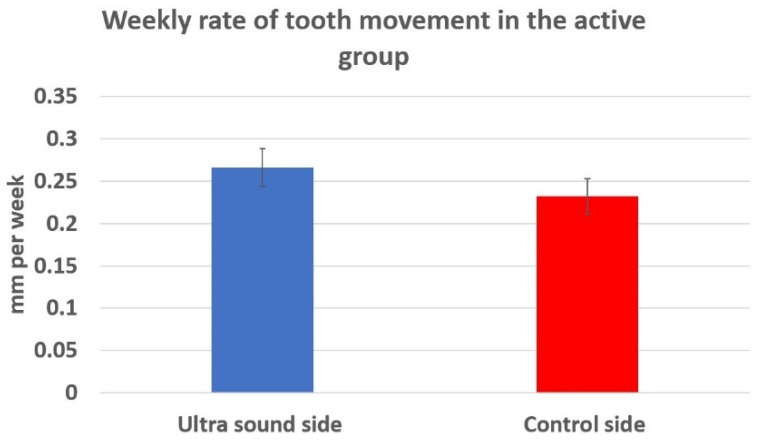
Comparison of the weekly rate of tooth movement in the active group.

**Figure 5 jcm-09-00804-f005:**
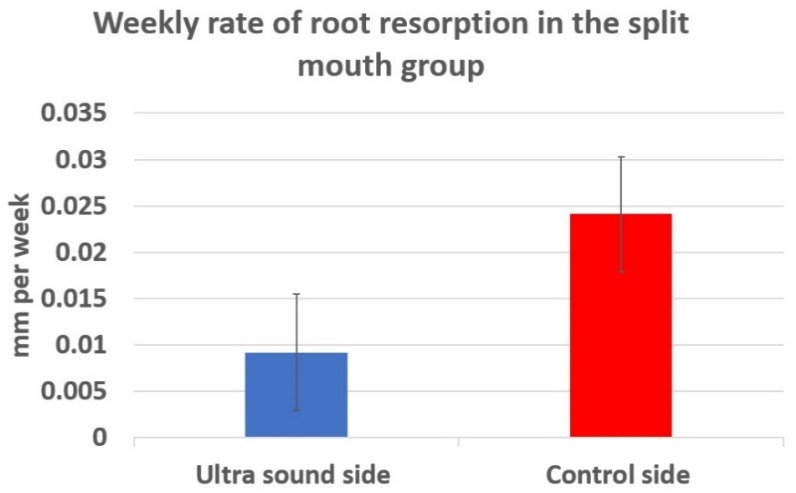
Comparison of the weekly rate of root resorption in the active group.

**Table 1 jcm-09-00804-t001:** Comparison of the weekly rate of tooth movement and root resorption between the positive and negative control groups.

	Tooth Movement			Root Resorption		
	Mean (mm/Week)	SD	*p* Value	Mean (mm/Week)	SD	*p* Value
Positive control	0.232	0.0855	0.11	0.0241	0.0226	0.32
Negative control (*n* = 10)	0.201	0.0398		0.02836	0.0247	

**Table 2 jcm-09-00804-t002:** Comparison of the weekly rate of tooth movement between the LIPUS and the positive control sides in the split-mouth group.

	Max (mm/Week)	Min (mm/Week)	Mean (mm/Week)	SD	Percent Change *	*p* Value
LIPUS (*n* = 17)	0.495	0.138	0.266	0.0927	29%	0.0164
Positive control (*n* = 17)	0.388	0.045	0.232	0.0855		

* As the study was a split mouth, the mean percent change for tooth movement rate (29%) was calculated as the mean of the individual percentage changes for the seventeen patients.

**Table 3 jcm-09-00804-t003:** Comparison of the weekly rate of root resorption between the LIPUS and positive control side in the split-mouth group.

	Max (mm/Week)	Min (mm/Week)	Mean (mm/Week)	SD	Percent Change *	*p* Value
LIPUS (*n* = 13)	0.49	−0.03	0.0092	0.0226	220.8%	0.0423
Positive control (*n* = 13)	0.057	−0.014	0.0241	0.0223		

* As the study was a split mouth, the mean percent change for root resorption rate (220.8%) was calculated as the mean of the individual percentage changes for the thirteen patients.

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
