# Peer review of "Effect of Low Intensity Pulsed Ultrasound (LIPUS) on Tooth Movement and Root Resorption: A Prospective Multi-Center Randomized Controlled Trial"

_jcm, 2020, doi:10.3390/jcm9030804_

Round 1

Reviewer 1 Report

The authors report a study on the effect of low-intensity pulsed ultrasound on tooth movement.  It is an interesting study and could be stronger if the sample size is bigger.  Although about 60 patients were recruited, but most of them are excluded from the final analysis.  Besides, it seems the samples were divided into more than two groups, I am not sure the results are still meaningful (e.g., the deviation is greater than the mean value).  The authors should address this.

In addition, I have several suggestions:

  1. Please specify how many patients are included in the tooth movement and root resorption analysis. It is clearer than knowing how many are excluded.
  2. Not sure how the 29% was obtained. The difference between 0.266 and 0.232 is not 29%.
  3. Full description should be given for LIPUS in the abstract.
  4. Superscript cm^2 should be used for the intensity.

Author Response

Reviewers’ comments. Responses /changes are in the tracked changed manuscript.

Reviewer 1:

Comments and Suggestions for Authors

The authors report a study on the effect of low-intensity pulsed ultrasound on tooth movement.  It is an interesting study and could be stronger if the sample size is bigger.  Although about 60 patients were recruited, but most of them are excluded from the final analysis.

Authors’ comment: The authors would like to thank the reviewer for the comment. Sixty patients were screened for the clinical trial out of whom only forty-seven participated. From the forty-seven, only twenty-one completed the trial with approved protocol and device usage of ≥67%. This part is now mentioned in the revised manuscript on page 3 line 131 – 133 as  well as Figure 1 was updated as well.

Besides, it seems the samples were divided into more than two groups, I am not sure the results are still meaningful (e.g., the deviation is greater than the mean value).  The authors should address this.

Authors’ response: We apologize about the confusion; There are two groups, one group was tested in split mouth study design. One side was randomly assigned as LIPUS and the other side was assigned as control. Group 2 was no device at all to avoid any possible contamination of LIPUS from the active to the placebo sides in group 1, group 2 was added with no ultrasound device. Details is in the revised manuscript Page abstract line 33-34 and page 4 line 153.

In addition, I have several suggestions:

  1. Please specify how many patients are included in the tooth movement and root resorption analysis. It is clearer than knowing how many are excluded.

Authors’ comment: The number of patients’ data analyzed in the tooth movement and root resorption is added in the revised manuscript. Please check Page 8 line 294 and Page 9 lines 310 – 312. Thank you.

  1. Not sure how the 29% was obtained. The difference between 0.266 and 0.232 is not 29%.

Authors’ comment: 29% was obtained by the average of individual percentages changes of seventeen patients. The explanation is now added to the revised manuscript on Page 8 lines 298 – 300. Thank you.

  1. Full description should be given for LIPUS in the abstract.

Authors’ comment: LIPUS description has been added to the revised manuscript on Page 1 lines 32, 35.

  1. Superscript cm^2 should be used for the intensity.

Authors’ comment: Thank you for the comment. ‘cm2’ is used in the revised manuscript on Page 1 line 37 and Page 2 line 89. 

Reviewer 2 Report

A manuscript “Effect of low intensity pulsed ultrasound on tooth movement:

a prospective multi-center randomized controlled trial,” has been reviewed.

This is a report studying on the effect of LIPUS stimulation during orthodontically canine retraction. The report showed LIPUS enhances the rate of orthodontic tooth movement and inhibit the root resorption. This is an interesting report expanding our understanding of the effect of LIPUS stimulation during orthodontic tooth movement. However, some revisions are necessary to improve the quality.

Major comments, 

Abstract

The authors should clearly state the study “outcome” in the Abstract.

Methods

The size of the loop used in Fig. 3 appears to be different between the left and right. Did the author used chart when made a loop and/or measured the force during loop activation?

How was the occlusal contact and/or occlusal force of the retract canines?

What was the ratio between the maxilla and the mandible in the subjects? It is quite different anchorage loss between maxilla and mandible. The measurement method in this study may affect the amount of tooth movement.

What do you think about this point?

 Discussion

The discussion is well written.

P10. L346”The statistically significant decrease~” This part has been cited incorrect reference. The reference number 60 may be correct. In any case, LIPUS enhances cementoblasts differentiation and prevent root resorption caused by odontoclasts is an illogical conclusion, and this mechanism will need further investigation.

Author Response

Reviewer 2:

Major comments,

Abstract

The authors should clearly state the study “outcome” in the Abstract.

Authors’ response. Thanks you, the outcome has been clearly stated in the abstract (line 40).

Methods

The size of the loop used in Fig. 3 appears to be different between the left and right. Did the author used chart when made a loop and/or measured the force during loop activation?

Authors’ response: We agree with the reviewer’s comment that the size of the loop was adjusted according to the distance between the bracket of the canine and mesial surface of the first molars’ tubes to equally deliver 100 gram force with moment to force ratio of 12 according to Burston’s segmented arch mechanics recommendation.  However to avoid confusion, a different patient’s photos are provided replacing the previous photos. Thank you.

How was the occlusal contact and/or occlusal force of the retract canines?

Authors’ response: Occlusal contact was minimized by using posterior bite turbos/bite ramps on the upper teeth. Thank you.

What was the ratio between the maxilla and the mandible in the subjects? It is quite different anchorage loss between maxilla and mandible. The measurement method in this study may affect the amount of tooth movement. What do you think about this point?

Authors’ comment: The ratio of maxilla and mandible in the subjects have been added to the revised manuscript on the Page 11 lines 348 – 353.

Discussion

The discussion is well written.

Authors’ response: Thank you.

P10. L346”The statistically significant decrease~” This part has been cited incorrect reference. The reference number 60 may be correct. In any case, LIPUS enhances cementoblasts differentiation and prevent root resorption caused by odontoclasts is an illogical conclusion, and this mechanism will need further investigation.

Authors’ comment: Authors thank the reviewer for correcting the reference. The right references and details have been added to the revised manuscript on Page 11 lines 378 – 383.  Thank you.

Reviewer 3 Report

Dear Authors

the article is interesting by the way some concerns have to be addressed before considering it for publication.

I made some suggestions to improve the manuscript.

I look forward to read the new version of the manuscript.

Kind regards 

Author Response

Reviewer 3:

  • The title of the article mentions the effect of low intensity pulsed ultrasound (LIPO) on tooth movements but in the description of the article the effect on root resorption is also considered.

Authors’ comment: “root resorption” has been added to the title of the revised manuscript. Thank you.

ABSTRACT

  • The subtitles are not present: Aim, materials and methods, results, discussion and conclusions.

Authors’ comment: Thank you for the suggestion, however, the authors followed the manuscript template available on the journal’s website and subheadings are not permitted.  

INTRODUCTION

  • According to the article: “The primary aim of the current study was to evaluate the effect of LIPUS on the rate of orthodontic tooth movement in the split-mouth clinical trial. The null hypotheses were: 1) There is no significant difference in the rate of tooth movement between LIPUS and control side; 2) There is no significant difference in the root resorption after orthodontic tooth movement between LIPUS and control sides. The alternate hypotheses are 1) LIPUS treated side will have accelerated tooth movement and reduced root resorption as compared to the control side.” why consider null and alternative hypotheses if the conclusion part does not respond to any of these hypotheses?

Authors’ comment: Thank you for the comment. The null and alternative hypotheses rejection and acceptance have been added to the revised manuscript on Page 12 lines 397 - 399

MATERIALS AND METHODS

  • According to the article: “The prospective patients were randomized into two groups. Group 1, in the split-mouth study, involved one side of the mouth receiving LIPUS treatment while the other side was inactivated and  served as a placebo or positive control… Group 2 was  composed of 10 patients that had no LIPUS treatment on either side, which served as a negative  control group and was included in the study (no blindness was applied to this group).” but in this investigation it was not determined how many people were included in each group.

Authors’ comment: Number of patients included in each group for tooth movement and root resorption has been added to the revised manuscript on Page 4 lines 138 – 142. Thank you.

RESULTS

  • According to the article: “ For the final analysis, there were twenty-one patients in the clinical trial analysis…with 5 male and 16 female subjects…Data from ten patients were collected and analyzed as a negative control group and were only compared to the positive control group to study if LIPUS was reaching to the positive control in the  active group and were not compared to the active side analysis… In the active treatment group, four patients were excluded due to the type of tooth movement being non-comparable… Six subjects from one trial site had dental radiographs instead of CBCTs, hence they were excluded from the root resorption analysis.” But if the present study determined that 21 people were selected but in this part of the article it’s mentioned that some patients were eliminated for different reasons, the collection of data shown in Tables 1,2 and 3 does not specify in the number of people who intervened.

Authors’ comment: Number of patients for tooth movement analysis has been added to the Table 2 on page 9, for root resorption to Table 3 on page 10. Thank you.

  • According to the article: “For the final analysis, there were twenty-one patients in the clinical trial analysis. The average age of twenty-one patients was 19.7 ± 6.63 (minimum = 12 years, and maximum = 37 years 5 months) with 5 male and 16 female subjects.” But can a sample be considered homogenous with a small group of 21 people where 16 people are female?

Authors’ comment: The authors do consider this as one of the limitation of the study and has been added to the revised manuscript on Page 11 lines 392 – 396. Thank you.

DISCUSSION

  • According to the article: ” This clinical investigation aimed to evaluate the effect of LIPUS on the rate of tooth movement and root length changes, as an indication of OITRR.” But in the part of the abstract it’s considered as aim only the evaluation of the effects of LIPUS on tooth movement.

Authors’ comment: The evaluation on root resorption has been added to the revised manuscript on Page 1 line 32. Thank you.

  • According to the article:” The primary limitation of this study is the sample size of twenty-one patients in a split-mouth design. In the future, a clinical trial with a larger sample size and clean treatment and control groups will be undertaken.” but having a reduced sample of 21 people, the conclusions obtained could be considered as scientific support in future research?

Authors’ comment: This study will help in sample size calculation from the standard deviation calculated and device usage compliance. This description is added to the revised manuscript on Page 11 line 392.

CONCLUSION

According to the article:” LIPUS increased the rate of tooth movement and decreased orthodontically induce root resorption when applied for 20 minutes per day for up to 6 months.”  but the conclusion is based on the favorable effect of LIPUS on movement and root resorption, then it would be necessary to consider in the title of this article the effect of LIPUS on root resorption.

Authors’ comment: “root resorption” has been added to the title of the revised manuscript.

Round 2

Reviewer 1 Report

I am still not sure the 29% is correct. 

since (0.266-0.232)/0.232= 14.6%

Unless it is calculated using other numbers, please specify.

Author Response

Sorry about the confusion, details about the calculation is attached in the enclosed file, please let us know if there is anything else needs to be clarify, thanks again.

Reviewer 2 Report

A revised manuscript “Effect of low intensity pulsed ultrasound on tooth movement: a prospective multi-center randomized controlled trial,” has been reviewed. Minor comments shown as below.

The picture in Figure 2 is unclear. Does the canine move this distance only 2 months?

Author Response

Does the reviewer mean (Figure 3)? yes, it is after 2 months, new revised caption is below and in the revised manuscript. Thank you for the comment.

Figure 3: A case in the LIPUS group at the beginning (top left) and 2 months of the treatment (Top right) the patient’s lower right side was treated by LIPUS and lower left side was control. CBCT scan showing root angulation after canine retraction (bottom left LIPUS treated side and right is control).